# Comprehensive Genome-Wide Approaches to Activity-Dependent Translational Control in Neurons

**DOI:** 10.3390/ijms21051592

**Published:** 2020-02-26

**Authors:** Han Kyoung Choe, Jun Cho

**Affiliations:** 1Department of Brain and Cognitive Sciences, Daegu Gyeongbuk Institute of Science and Technology (DGIST), Daegu 42988, Korea; 2Department of Biomedical Science and Engineering, Gwangju Institute of Science and Technology (GIST), Gwangju 61005, Korea

**Keywords:** translational control, activity-dependent gene expression, mTOR signaling pathway, local translation, phosphoTRAP, ribosome profiling, RiboTag, Autism spectrum disorder, depression

## Abstract

Activity-dependent regulation of gene expression is critical in experience-mediated changes in the brain. Although less appreciated than transcriptional control, translational control is a crucial regulatory step of activity-mediated gene expression in physiological and pathological conditions. In the first part of this review, we overview evidence demonstrating the importance of translational controls under the context of synaptic plasticity as well as learning and memory. Then, molecular mechanisms underlying the translational control, including post-translational modifications of translation factors, mTOR signaling pathway, and local translation, are explored. We also summarize how activity-dependent translational regulation is associated with neurodevelopmental and psychiatric disorders, such as autism spectrum disorder and depression. In the second part, we highlight how recent application of high-throughput sequencing techniques has added insight into genome-wide studies on translational regulation of neuronal genes. Sequencing-based strategies to identify molecular signatures of the active neuronal population responding to a specific stimulus are discussed. Overall, this review aims to highlight the implication of translational control for neuronal gene regulation and functions of the brain and to suggest prospects provided by the leading-edge techniques to study yet-unappreciated translational regulation in the nervous system.

## 1. The Emerging Importance of Translational Regulation in Neuronal Gene Expression

Translation is an essential process of gene expression where ribosomes synthesize polypeptide chains that will be folded into functional proteins by the guidance of mRNA templates. Translation demands a large amount of various cellular resources, including ATPs and amino acids to synthesize one polypeptide chain, that is to say, that it is one of the most costly processes energetically [1]. Hence, the translation should be sophisticatedly controlled in any type of cell or tissue, otherwise it may not sustain their viability as well as function. Indeed, compromised translation caused by mutations in ribosomal or other proteins that constitute translational machinery resulted in a wide range of diseases such as anemia, developmental deficiencies, and neurodegeneration [2,3,4].

One piece of evidence that supports the importance of translational control for cellular and organismal processes is ribosomopathies, a group of human disorders that are attributed to mutations in specific ribosomal proteins or ribosome biogenesis factors [5,6]. Notably, the phenotype patterns of ribosomopathies are heterogeneous, and in many cases tissue-specific, in spite of the omnipresent requirement of ribosomes in all cells and tissues. As a representative example, Diamond-Blackfan Anemia (DBA) that is caused by heterozygous loss-of-function mutations in one of 18 ribosomal protein-coding genes, resulted in haploinsufficiency of ribosomal proteins in multiple lineages of hematopoietic cells, but only erythroid cells were selectively ablated [7]. Subsequent studies revealed that the diminished ribosomal proteins led to the reduced translation of specific mRNAs such as GATA1 mRNA that can cause erythroid depletion [8]. Although the mechanism underlying the selective inhibition of translation on particular mRNAs in DBA patients remains to be elucidated, the tissue- and transcript-selective phenotype of the ribosomopathy supports the significance of tight controls of translational processes.

Adverse effects caused by an abnormality of the translational process are exemplified by other diseases as well as ribosomopathies. The Wolcott–Rallison Syndrome (WRS) is associated with mutations in the catalytic domain PERK, eukaryotic initiation factor 2 alpha (eIF2α) kinase, which results in inappropriate activation eIF2α during unfolded protein response [9]. The abnormally enhanced translation induces accumulation of unfolded proteins, leading to irreversible damages to pancreatic β-cells, loss of them, and hence the onset of diabetes. The specificity of mRNAs that are abnormally translated in WRS has not been thoroughly studied. However, in squamous cell carcinoma, eIF2α is known to potentiate translation of many oncogenic transcripts in a non-conventional 5′ untranslated region (UTR)-dependent manner, alluding to the target specificity of translational activation by the protein [10].

Multiple neurological disorders have also been linked to the anomaly of translational controls. The persistent phosphorylation of eIF2α was also observed in neurodegenerative diseases, including Alzheimer’s disease and Parkinson’s disease [11,12]. The activated eIF2α induces translational activation of a subset of mRNAs such as BACE1 and ATF4 that are involved in the onset of Alzheimer’s disease and the inhibition of eIF2α activity reduced accumulation of misfolded proteins, ameliorating neurodegeneration in Parkinson’s disease. An inherited brain disease, Leukoencephalopathy with vanishing White Matter (VWM), has been linked to mutations in genes encoding eIF2B, the guanine-nucleotide-exchange factor that is involved in energy expenditure during translational initiation [13]. These mutations in enzymatic and regulatory subunits of eIF2B were in some cases known, in other cases suspected to disturb eIF2 function, suggesting the disrupted translation is the most likely cause of VWM in spite of yet-unappreciated mRNA specificity. These etiological studies revealed how important tight control of the translational process is for the development and homeostasis of neuronal tissues.

The loss of controls in translational processes can result in turmoils of neuronal activities as well that may cause neuropsychiatric diseases such as autism disorders. One pronounced example is Fragile X Syndrome (FXS), the most common form of inherited mental retardation. FXS is caused by a loss of an RNA binding protein Fragile X Mental Retardation Protein (FMRP) that is associated with actively translating poly-ribosomes [14]. FMRP binds a specific subset of mRNAs encoding synaptic proteins and is involved in the transport of them to post-synaptic sites from the nucleus and protein synthesis of the mRNA upon synaptic activation. It has been demonstrated that the abrogated translation of FMRP-associated mRNA is the most likely cause of FXS [15]. The studies on the function of FMRP and its association with FXS provide strong evidence supporting that translational regulation that occurs in response to neuronal activation is crucial for normal activity of the brain. Indeed, several mechanisms and effects of translation controls that occur in neurons in activity-dependent manners have been reported over the past decades [16,17,18,19,20]. However, activity-dependent translational controls are less thoroughly studied compared to transcriptional ones due to the limitation of methods to study them. Notably, recently developed methods mainly derived from RNA sequencing revolutionized how we can access the content of translatome, thereby opening a new window to comprehend the role of translational control in response to neuronal activity. Here we review the precedent and recent studies on this subject focusing on the experimental techniques and try to suggest perspectives for future directions.

## 2. Translational Control Bridging Neuronal Activation and Genetic Program

Neuron utilizes the changes in membrane potential to compute incoming signal and to transmit information. While this dynamic form of the electric event lasts only for slightly longer than a millisecond, the neuronal activity can leave a molecular trace on the neuron by eliciting a change in gene expression, so-called immediate early gene response. In addition, a certain pattern of electric firing that leads to a long-term change in interneuronal communication—synaptic plasticity—leads to molecular modifications of neuronal structure and function that underlies potentiated or depressed synaptic transmission. The interaction between the electric processing of information and the molecular basis of cellular component enables dynamic adaptation to the external world while preserving the identity of an individual.

There has been extensive effort to understand how neural activity transforms gene expression program. Substantial evidence indicates that transcriptional regulation underlies activity-dependent gene expression, which has been excellently reviewed elsewhere [21,22]. Very briefly, the activation of membrane channels induced by synaptic transmission leads to an increase in cytosolic calcium levels. Subsequently, Ca^2+^-dependent kinase cascade leading to the activation of the downstream signaling pathway induces the activity of transcriptional factors. The activation of the signaling pathway that regulates the transcription can have a long-lasting effect on the transcription of the neuron by functioning in concert with epigenetic modifiers. 

In addition to transcriptional regulation, translational control constitutes another critical part of the regulation of the protein level in response to neural activity. Long-term synaptic plasticity induced by a specific form of neural activation provided evidence that a neuronal electric signal that can induce persistent change lasting for a few days requires protein synthesis as shown in classic studies in *Aplysia* [18]. Pharmacological studies using translational blockers, such as anisomycin and puromycin, in rodents were instrumental in the fortification of the view that protein synthesis is a crucial step in learning and memory as well as long-term potentiation [16,23]. Pharmacological loss-of-function study with a possible side effect was later complemented by fluorescent reporter-based time-lapse imaging and histological investigation that allowed the observation of translational progress upon neural stimulation [19,20]. Recent technological progress in real-time single molecule biophysics enabled direct monitoring of translational regulation in response to neural activity [24], further highlighting the significant contribution of translational control in the activity-dependent modification of gene expression. 

Although transcriptional control and translational control together comprises critical steps in the regulation of protein level, multiple reports indicating low correlation of transcriptome and proteome [25,26,27] suggest that translational control may function as an independent module in activity-dependent gene expression control and that neuronal translational control deserves attention at least as much as transcriptional control of the nervous system.

## 3. Mechanisms Underlying Activity-Dependent Translational Control

In contrast to neural activity-elicited modulation in the transcription of which mechanisms result in the nuclear event, the activity-dependent translation link membrane event to the translational machinery in the cytosol, whether it be dendrite, soma, or axon. The relatively short distance from the neural membrane to the site of translational control in combination with the compartmentalization of neuronal morphology endows a great amount of flexibility and rapidness to translational control. For example, translational control can shift the profile of translatome within 5 min [17]. The effect of translational control can be refined to a small neuronal structure such as a dendritic shaft or synaptic bouton [19,20]. In the following subsections, we will explore what has been elucidated regarding cellular mediators of activity-dependent translational controls in neurons (Figure 1).

### 3.1. Kinase Pathway Modifying Translation Factors

Eukaryotic translation machinery synthesizes protein by forming peptide bonds between amino acids dictated by the information contained in mRNA in three steps—initiation, elongation, and termination. Recent understanding of molecular machinery controlling eukaryotic translational machinery is insightfully reviewed in detail by Hershey, Sonenberg, and Mathews [28] as well as other review articles of the current Special Issue.

Briefly, important regulatory procedures can be found first at the initiation. Phosphorylation of eIF2 plays an important for the control of initiation by reducing translational activity in general while positively regulating translation of mRNA containing an upstream open reading frame [16]. The eIF2α in neurons is mainly phosphorylated by general control nonderepressible 2 (GCN2) and double-stranded RNA-dependent protein kinase (PKR), which is induced by patterned neuronal activation and the treatment of neurotrophic factor in addition to behavioral tasks including learning [29]. The reduction of eIF2α phosphorylation in mutant mice lacking GCN2 enhanced the formation of long-term plasticity and learning-dependent behavioral tasks. On the other hand, upregulated eIF2α phosphorylation in conditional overexpression of PKR driven by CaMKII promoter reduced the maintenance of long-term potentiation (LTP) and impaired the expression of learned behavior such as tone-paired freezing [30].

Next, post-translational modifications of elongation factors are also crucial in activity-dependent translational control. Eukaryotic elongation factor 2 (eEF2) is required for the progression of translation, while phosphorylation of eEF2 by eukaryotic elongation factor 2 kinase (eEF2K, also known as calmodulin-dependent protein kinase III (CaMKIII)) inhibits the elongation of translation [31]. An increase in intracellular calcium level mediated by the N-methyl-D-aspartate (NMDA) glutamate receptor can increase the activity of eEF2K to phosphorylate eEF2, subsequently suppressing translation in general. The importance of eEF2 in activity-dependent translational control is demonstrated in the example of Arc protein synthesis in mGluR-dependent long-term depression (LTD) [32].

### 3.2. mTOR Signaling Pathway

Mammalian target of rapamycin (mTOR, also referred to as mechanistic target of rapamycin) signaling pathway plays a key role in translational regulation elicited by neuronal activation [33]. Generally, the mTOR signaling pathway is critical in cell signaling pathway controlling protein synthesis, cell death, cell growth, metabolism, and autophagy [34,35]. In neurons, regulation of the mTOR signaling pathway is implicated in brain development, synaptic plasticity, and circadian rhythm. Mutations and dysregulation in the component of the mTOR pathway are suggested to be involved in several neurological diseases and psychiatric disorders [33,36]. Serine/threonine protein kinase mTOR can form two complexes, mTORC1 and mTORC2, based on binding partners. Among them, mTORC1 is mainly linked to neuronal events such as regulating neuronal excitability as well as learning and memory [37,38]. Once mTORC is activated, it leads to the modification of the downstream signaling pathway leading to the translational regulators including 4E-BPs (4E-BP1, 4E-BP2, and 4E-BP3) and ribosomal protein S6 kinase beta-1 (S6K1). Phosphorylation of 4E-BPs results in dissociation of 4E-BPs with eIF-4E results in the initiation of translation [34].

Long-term synaptic plasticity offered a platform to understand the role of the neuronal mTOR signaling pathway in response to neuronal activity. A pioneering study by the Schuman group reported that mTOR is localized in the postsynaptic compartment of the dendritic shaft of hippocampal primary neurons and that rapamycin can block long-term potentiation (LTP) elicited by either high-frequency stimulation or brain-derived neurotrophic factor (BDNF) treatment [39]. The effect of rapamycin was crucial during the induction phase of LTP when postsynaptic responses to electric stimulation were increasing [40]. In response to repeated high-frequency stimulation causing LTP, the phosphorylation of S6K1 was observed in dendrite in a phosphoinositide 3-kinase (PI3K)-dependent manner. The effect of rapamycin in long-term synaptic plasticity was also evident in vivo. Spatial memory probed by the Morris water maze is associated with the long-term potentiation of the hippocampal region. The injection of rapamycin into the dorsal hippocampus impaired spatial memory of rats [41]. Spatial memory, as well as phosphorylation of the mTOR downstream targets including S6K1 and 4E-BP1, was increased by intrahippocampal glucose administration and ameliorated by the treatment of 5-aminoimidazole-4-carboxamide ribonucleotide (AICAR), a 5’ AMP-activated protein kinase (AMPK) activator. Interestingly, the genetic reduction the S6K1 level, which increased in Alzheimer’s disease (AD) patients, in the AD mice model was capable of rescuing the loss of spatial memory [42]. Synaptic plasticity in the amygdala mediates auditory fear conditioning that pairs an auditory cue, as a conditioned stimulus, to electric shock, as an unconditioned stimulus, to elicit a learned fear response to a previously neutral auditory cue [43,44]. Microinfusion of rapamycin into the amygdala decreased freezing fear response to an auditory cue with a correlated decrease in the phosphorylation level of S6K1 [45].

Accumulating evidence that indicates the role of the mTOR signaling pathway in long-term potentiation fueled the search to understand the role of mTOR signaling pathway and rapamycin in the synaptic plasticity-mediated clinical model such as the reinstatement of drug addiction and alcohol abuse [46]. Indeed, microinjection of rapamycin into the nucleus accumbens core suppressed relapsed drug-seeking behavior in rats [47]. Systemic pharmacological suppression of the mTOR pathway also reduced the reinstatement of alcohol-seeking behavior [48]. In both cases, decreasing relapsed addictive behavior by local or systemic inhibition of the mTOR signaling pathway was not simply due to the induction of negative emotional response. Based on this evidence in rodents, a phase 1 clinical trial to repurpose rapamycin to treat alcohol use disorder is ongoing (ClinicalTrials.gov identifier: NCT03732248).

The activity of mTORC1 is mainly regulated by signaling cascade including PI3K, AKT/protein kinase B, and tuberous sclerosis complex (TSC) [34,35]. It is well established that neurotrophic factor signaling such as BDNF and nerve growth factor activates this pathway [49,50]. MAPK pathway also participates in neurotrophic factor-mediated regulation of mTOR signaling pathway through mTOR complex 2 [51]. PI3K-AKT pathway can also be activated by the activation of the metabotropic glutamatergic receptor, mGluR5, associated with Homer [52,53]. Still, understanding why neuronal activity leads to regulation of mTOR signaling pathway requires further investigation. In this regard, it is interesting that direct current stimulation of the hippocampus *ex vivo* leads to synaptic plasticity, which is dependent upon mGluR5 and mTOR signaling pathway [54], suggesting that activity-dependent regulation of mTOR signaling pathway may involve molecular mechanisms which underlie other regulatory contexts. Additionally, as the mTOR signaling pathway is involved in a variety of cellular contexts, interrogating the contribution of mTOR signaling pathway-mediated translation in related pathophysiology will provide a fundamental basic and clinical insight.

### 3.3. Local Translation

A neuron is a very specialized cell in its morphology and functions. Information from upstream, presynaptic neurons as a form of the electric signal reaches the dendrite of a neuron through chemical transmission across the synaptic cleft. The dendrite is enriched with proteins that function to detect chemical signals released from the presynaptic terminal and to modulate the physical structure and chemical composition of the postsynaptic compartment. The soma of a neuron contains the nucleus of the cell and integrates synaptic inputs from several branches of dendrites. If temporally and spatially integrated signal given to a neuron is over the threshold, the action potential is generated at the axon hillock to travel along the axon. The axon has special cellular machinery to efficiently transmit the action potential and has ion channels and secretory machinery at its terminal to transduce the neural impulse to downstream neurons. As each compartment of neurons perform specialized functions utilizing locally enriched proteins, proper localization of the protein to dendrite, soma, or axon and cellular machinery to establish and to maintain the protein localization are crucial in the normal functioning of neurons. In many subcellular organelles including the nucleus, mitochondria, and endoplasmic reticulum, signal sequence within polypeptide chain plays an important role in the localization of the protein to its cellular target site. However, the consensus signal sequence for the subcellular compartment of neurons has not been discovered yet. The localization of mRNA to a specific subcellular location and local translation of the mRNA are important for the localization of proteins to the neuronal subcellular compartments such as dendrite and axon [55,56,57].

Erin Schuman’s group provided pioneering evidence that local translation is under the regulation of synaptic activity [20]. Utilizing time-lapse imaging of the expression of green fluorescent protein (GFP) in hippocampal neurons, the authors monitored the local changes in GFP fluorescent intensity, which is determined by synthesis and degradation. Pharmacological blockade of the action potential as well as miniature excitatory synaptic current lead to increased expression of GFP. The increased reporter activity was disproportionately distributed along the length of dendrites, suggesting that local translational is under the regulation of synaptic events. Miniature excitatory synaptic current-elicited local translation increased the surface expression level of GluR1 protein at a certain point of dendrite and facilitate homeostatic synaptic plasticity by increasing the amplitude of AMPA receptor-mediated spontaneous current [19]. As homeostatic synaptic plasticity is considered to maintain the overall tone of synaptic transmission, activity-dependent local translational can control and stabilize neural computation in the long term. These experiments demonstrate that neuronal activity can modulate translational machinery governing protein synthesis in a finely defined cellular compartment and that activity-dependent translational control provides adaptive value to neural computation.

What are the cellular mechanisms mediating local regulation of translation upon neuronal activity? Although it still remains largely unknown, pioneering studies initiated the interrogation of candidate mechanisms underlying activity-dependent local translational control. The first candidate is a localized maturation of micro RNA in the vicinity of neuronal excitation [58]. Nanostring analysis of the physically dissected hippocampus identified the expression of several hundreds of miRNAs in the neuropil. A precursor of miR-181a, one of the most abundant miRNAs in the neuropil, was identified in both soma and dendrite. The authors then employed a fluorescent reporter that can reflect the maturation of miR-181a and monitored its real-time dynamics upon local neuronal excitation by using two-photon glutamate uncaging. The glutamate uncaging has been widely utilized in neuroscience, specifically by synapse neuroscientists, to examine the effect of local stimulation typically limited within several micrometer ranges which is equivalent to the diameter of a single synapse [59]. Notably, the uncaging induced the maturation of miR-181a at a single-synapse level within 20 s, which was associated with the reduced translation of CaMKIIa, a target of miR-181a. Another possibility is a locally regulated granule formation through phase separation. FMRP, which we will cover in detail in the following section, is an RNA-binding protein that generally represses mRNA translation. The FMRP-mediated translational control can be reversibly regulated through phosphorylation by S6K1, casein kinase 2 (CK2), and protein phosphatase 2A (PP2A) [60,61,62]. It is notable that phospho-FMRP is likely to form a locally restricted lipid droplet (neurogranule) serving as a repressive complex containing ribosome and mRNA [63].

Considering that the compartment of dendrites can function as an independent unit of neural computation and integration [64], activity-dependent local translational regulation may play a more important role than currently appreciated, endowing neurons to record the trace of a recent neural event for fine tuning of repeated computational need. Employing leading-edge neuroscientific tools, such as finely controlled local stimulation employing optogenetics and patterned illumination, would be highly beneficial in further understanding the biological role and mechanisms of activity-dependent local translation.

## 4. Translational Control in Neurodevelopmental and Psychiatric Disorders

The importance of translational control in the pathophysiology of the brain is widely appreciated. Here we will explore the recent advances in understanding the contribution of abnormal translational regulation in neurodevelopmental and psychiatric disorders, focusing on autism spectrum disorder and depression. For the implication of translational machinery on neurological disorder, an excellent review is available elsewhere [33].

### 4.1. Autism Spectrum Disorder

Autism spectrum disorder (ASD) is a debilitating neurodevelopmental disorder with core symptoms of (1) abnormal social functioning and (2) repetitive and restricted set of behaviors, based on the diagnostic criteria in Diagnostic and Statistical Manual of Mental Disorder, 5^th^ edition (DSM-5) [65]. Growing evidence indicates that ASD is heavily affected by genetic factors. It is noteworthy that mRNA translation-associated factors are found among the ASD-associated genes [66].

One of the notable cases is fragile X syndrome (FXS), a subset of genetic ASD, caused by a lack of functional expression of FMRP, encoded by Fmr1. FMRP is an RNA-binding protein that is abundantly found in neuronal granule [67,68]. FMRP appears to participate in multiple steps of post-transcriptional and translational procedures including pre-mRNA processing, neuronal granule transport, translational suppression. In addition to the earlier mentioned role of FMRP in activity-dependent local translational control, it is involved in mGluR5-mediated LTD that leads to repressed translation of several target mRNA including Arc [32,69]. It is noteworthy that the translation of FMRP itself is controlled by neuronal activation [70], possibly forming a translational positive feedback circuit. Although it still remains unclear how loss-of-function mutation of FMRP leads to the pathogenesis of ASD, it is of clinical interest that counteracting the cellular procedure caused by mutation of FMRP rescued ASD-like behavioral symptoms in animal models of ASD [67]. For example, pharmacological inhibition of mGluR5 in the FXS model rescued imbalanced protein synthesis and synaptic function [71,72]. There are currently more than ten clinical trials ongoing for pharmacological intervention of FXS.

Another important gene in ASD-related translational control is TSC (tuberous sclerosis complex). Mutation in TSC complex component, either TSC1 or TSC2, increase the penetrance of ASD [73]. TSC complex, which is inhibited by AKT-mediated phosphorylation, regulates the activity of mTOR complex through suppressing Ras homolog enriched in brain (RHEB). Indeed, haplodeficiency of TSCs leads to hyperactivation of mTOR complex and excessive synthesis of protein [74]. Pharmacological intervention targeting mTOR complex yielded promising results in TSC patients [75].

Understanding whether there is a core set of FMRP target genes causing neurodevelopmental deficit in ASD or whether the imbalance in global, dendritic, and synaptic proteome trigger abnormal development of social functioning should be critical future goals. Considering the development of the nervous system requires finely orchestrated interplay between the sequential manifestation of the genetic program and the neural activity of developing neural circuits, comprehension of altered activity-dependent translation in patients and ASD animal models would be of great importance to develop mechanism-based therapeutics to ASD.

### 4.2. Depression

Recent progress in understanding the action of antidepressant fluoxetine and ketamine suggests a long-term modification of neural structure and function as critical components of their therapeutic effects (Figure 2). Notably, several studies indicate that the mTOR signaling pathway, a key player in translational control, is involved in the pharmacological outcome of antidepressants, fluoxetine, and ketamine. Fluoxetine, a selective serotonin reuptake inhibitor (SSRI), has long been known for its antidepressant effect. In mice, chronic stress has widely been used as a model of depression, leading to several depression-like symptoms including anhedonia, anxiety, and helplessness. Chronic treatment of fluoxetine can recover anhedonia in the chronically stressed mice, where the co-treatment of rapamycin with fluoxetine abolished the recovery effect [76]. Along with behavioral recovery, it is notable that phosphorylation levels of mTOR target molecules, S6K1 and 4E-BP1, were regulated similarly. Phosphorylation levels decreased in depression-like animals and were rescued by chronic fluoxetine treatment in an mTOR signaling-dependent manner.

Accumulating clinical evidence demonstrates that ketamine, an antagonist of the NMDA receptor, can rapidly alleviate depressive disorder at a subanesthetic dose [77,78]. Monteggia, Kavalali, and colleagues suggested translational mechanisms underlying ketamine-induced anti-depressant action [79]. Antidepressant effect elicited by ketamine treatment was blocked by anisomycin, suggesting the contribution of protein synthesis. The authors further discovered that ketamine treatment increases the expression of BDNF with a decreased level of phosphorylated eEF2 without a noticeable change in phospho-mTOR levels. Importantly, pharmacological inhibition of eEF2K phenocopied ketamine-induced antidepressant action in a BDNF-dependent manner. Another antidepressant model with enhanced GABAergic signaling by employing somatostatin-positive neuron-specific conditional deletion of GABA_A_R gamma subunit consistently indicates decreased phosphorylation of eEF2, although the translational outcome in this model was unclear [80]. It is also of note that the mTOR signaling pathway may play a role in ketamine-induced changes, such as the lengthening of dendrite length and soma size of primary dopaminergic neurons BDNF and AMPA receptor signaling [81].

As we explored in this section, the significance of translational control in psychiatric disorders is emerging. Although accumulating evidence solidly indicates the role of translation machinery in the pathology of psychiatric disorder and its recovery, it still remains unclear what protein is instrumental in mediating the therapeutic effect and which signaling molecule is the best molecular target for pharmacological intervention to optimally yield the safety and efficacy of therapeutics. Genome-wide unbiased technologies to understand translational control in neurons, which we will explore in the following section, offers a promising and exciting approach to obtain clues for the development of mechanism-based therapy for ASD and depression in addition to other psychiatric disorders.

## 5. Genome-Wide Approaches to Study Translational Controls in Neurons

The early studies on gene expression controls had relied on conventional methods that selectively amplify signals from the molecular behaviors of one or a few specific proteins and nucleic acids in a “targeted” and “biased” manner. One caveat of these approaches is that they can become time-consuming and labor-intensive when multiple target candidates should be examined. For the past decades, the development and application of techniques to estimate the number of thousands of biological molecules within one experiment such as microarray and mass-spectrometry have enabled “comprehensive” and “unbiased” assessment of molecular actions of nucleic acids and proteins. Moreover, the recently developed methods based on high-throughput sequencing technique has broadened our coverage to study post-transcriptional events as well as transcriptional ones at a genome-wide scale. Here, we review the advance of the techniques that have added insight into genome-wide studies on translational regulation on neuronal genes (Figure 3).

### 5.1. Polysome Profiling

The microarray was originally developed to estimate the expression of thousands of genes at a genome-wide level, i.e., the steady-state amounts of mRNAs encoded by the genes [82]. The technique has contributed to the identification of unknown differentially expressed genes in a wide range of biological processes and provided important clues to study the regulatory mechanisms underlying the expression controls [83]. However, the applications of microarray have been restricted mostly to study the genes that exhibit the alteration in transcription and mRNA stability, rather than translation. The limitation of the technique was attributed to its way to detect gene expression level that is dependent on the hybridization of the probes that are complementary to the pre-defined partial sequences of mRNAs. The hybridization-based protocol restricts the detection or distinguishment of partial or whole transcripts that the probes hardly recognize and may be enriched or depleted during experimental treatment to isolate translational machinery. In spite of the limitation, several studies exploited microarray combined with poly-ribosome (polysome) fractionation that is called polysome profiling to estimate translational activities of mRNAs in neurons as well as other types of cells [84].

Polysome profiling is based on the assumption that transcripts associated with polysome are being actively translated in general. Ribosomal subunits, monosome, and polysome are separated by density-mediated sedimentation through sucrose gradient. The mRNAs sequestered in high-density fractions where heavy polysome resides harbor high translation activities and those found in low-density fractions where monosome and light polysome have poor or mediocre translation. Two studies exploited polysome profiling to uncover selective translational controls in the neural context. Brown et al. identified a specific set of neuronal mRNAs that exhibited abnormal polysome profiles in fragile X cells and interacted with the FMRP ribonucleoprotein complex to define the target transcripts of the RNA binding protein [14]. Although they suggested putative targets of FMRP-mediated translational regulation, only fifty percent of FMRP-associated mRNAs confirmed by immunoprecipitation showed alteration in polysome profiling. This discrepancy might be attributed to the aforementioned caveat of the microarray-based protocol. Another study using polysome profiling revealed BDNF regulates the translation of specific mRNAs in Rapamycin-PI3K dependent manner during neural development [85]. Schratt et al. isolated polysome-associated mRNAs in mature and immature neurons and identified the transcripts that showed BDNF-induced and rapamycin-restricted enrichment in polysome. These studies demonstrated the power of genome-wide approaches that provide clues to unveil yet-unappreciated translational controls by scrutinization of the transcriptome, not just a few numbers of representative transcripts. However, polysome profiling has several limitations that originated from polysome isolation as well as the microarray method. First, fractionation using sucrose gradient sedimentation is too arbitrary and approximate to preserve the quantitative information of the translational change of all mRNAs. For example, mRNAs whose distribution changed within the same fraction cannot be identified by this technique. Furthermore, the pivotal assumption of polysome profiling that mRNAs associated with multiple ribosomes are actively translated is not always valid. Stalled ribosomes as well as active ones move to a heavy fraction of polysome during sucrose gradient sedimentation, suggesting that polysome profiling cannot perfectly determine more- and less-translationally active mRNAs [86]. These pitfalls of polysome profiling and microarray can be, at least partially, supplemented by the recently developed method exploiting next-generation sequencing technique, ribosome profiling.

### 5.2. Ribosome Profiling

High-throughput sequencing (also called next-generation sequencing) allows one to determine a large collection of DNA molecules with millions of base pairs in a massive and parallel manner within one experiment. Alignment of the sequences of individual DNAs that originate from chromosomal DNAs or RNAs onto genome or transcriptome provides quantitative as well as qualitative information of the original molecules. High-throughput sequencing does not have the technical issue that comes from the hybridization with the pre-defined probe set in microarray, providing more versatile applications that scrutinize the information of the DNAs or RNAs associated with many biomolecular complexes [87].

One of the applications that are called ribosome profiling or Ribo-seq revolutionized studies on translational controls. Previously, many studies that conjectured and found the target mRNAs of translational regulation had relied on the comparison between the changes in the steady-state level of the mRNAs and proteins due to the lack of any quantitative method to directly measure protein synthesis rates. Although polysome profiling provides numerical estimates that globally reflect protein synthesis rates, it has clear limitations as discussed. To measure the translation rate of individual transcripts at a genome-wide level, Weissman’s group developed ribosome profiling [88,89]. The technique relies on the isolation of ribosome footprints that are protected by ribosome upon nuclease digestion. In detail, nuclease treatment degrades all mRNA regions that are not protected by ribosomes and the protected RNA fragments of ~30 nucleotides (nt) length, ribosome footprints, present where ribosomes are located, i.e., where translation takes place. Individual ribosome-ribosome footprint complexes are isolated by sucrose-mediated sedimentation or gel filtration, which is followed by gel-electrophoresis-mediated size selection and rRNA removal to purify the ribosome footprints. The sequences of the purified ribosome footprints are determined by high-throughput sequencing, which defines the in vivo positions of ribosomes that translate mRNAs at nucleotide level resolution. The alignment of ribosome profiling data onto transcriptome manifests two features that support it successfully detecting in vivo actions of ribosomes: three-nucleotide periodicity representing translocation of ribosomes by codon and coding-DNA-sequence (CDS) enrichment indicating the concentration of ribosomes in protein-coding regions.

The first study utilizing ribosome profiling demonstrated the power of the technique to examine genome-wide translation in eukaryotes. Ingolia et al. performed ribosome profiling paired with RNA-seq to calculate translational efficiencies of all individual mRNAs that are defined by the ratio of the normalized read counts of the ribosome profiling to those of the RNA-seq [88]. Translation efficiencies may be a much better measure to quantitatively evaluate translational activities of transcripts given that ribosome profiling does not have the two aforementioned issues polysome profiling has. The quantitative power of ribosome profiling to estimate genome-wide gene expression was supported by the observation that the read counts of individual mRNAs from ribosome profiling vis-a-vis RNA-seq showed a better correlation with the amount of proteins measured by mass-spectrometry. In addition to its strength as a tool to evaluate protein synthesis rates, ribosome profiling enables to determine the precise location of ribosome association that can lead to the discovery of novel coding region or regulatory element of translation. One representative example is the determination of the upstream open reading frame (uORF) by the technique. The studies by Ingolia et al. and other groups applied ribosome profiling to estimate the translation of putative uORFs and adjacent CDSs to them in a wide range of contexts, elucidating their positive or negative roles in translation of the protein-coding regions [88,90,91,92].

By virtue of its potency to quantify gene-specific translation, ribosome profiling has been exploited to examine translational controls in diverse types of cells and tissues [93,94,95,96]. One impressive example that nicely shows the relevance of ribosome profiling as a measure of protein synthesis is the studies on the proportional synthesis of polycistronic transcripts. Weissman’s and Barkan’s groups revealed that proteins constituting multimeric complexes are synthesized from a single polycistronic mRNA proportionally to their stoichiometry ratio within the assemblies in bacteria and chloroplasts, respectively, demonstrating the quantitative precision of this technique [97,98]. The first study using ribosome profiling to examine translational regulation in the nervous tissue was performed by Cho et al. [17]. They analyzed paired translatomes and transcriptomes of mouse hippocampus along the time course of long-term memory formation, revealing three different types of repressive gene regulation that are coordinated by translational and transcriptional controls at the steady-state, early and late-phase. Their study supports that early regulation at the translational level of specific genes such as Nrsn1 (Neurensin) may be relevant for memory consolidation, though it was unclear whether the regulation is neuron-specific or not due to the heterogeneity of the tissue. Another impressive finding in their study was persistent translational repression of ribosomal protein-coding mRNAs in the hippocampus compared to mouse embryonic stem cells, which is recapitulated in more neuron-specific context by another group. Blair et al. applied multiple high-throughput sequencing techniques including ribosome profiling to human embryonic stem cells and neuronal cultures, showing suppression of ribosomal protein expression at translation level in the differentiated neurons that is likely to be coordinated by mTOR signaling [99]. On the other hand, ribosome profiling has been exploited by ribosome profiling specifically to neural stem cells or neurons in diverse neuronal disease models such as fragile X syndrome, and focal malformations of cortical development to identify gene-specific translational controls that are associated with the onset of the disease [100,101].

One major hurdle to apply ribosome profiling to study activity-dependent neuronal translation controls that occur in vivo tissue is, as shown in Cho et al.’s study and pointed out by Mathew et al., how to isolate neurons that are responding to and activated by external stimuli from the entire brain tissues [17,102,103]. Combining the ribosome profiling technique with TRAP or RiboTag that will be discussed later may provide a solution to overcome the hurdle. One recent study isolated ribosomes selectively from the excitatory neurons by using Camk2a-RiboTag mice and performed ribosome profiling, providing excitatory neuron-specific translatome [104]. However, this approach still lacks a way to construct the cell type-specific transcriptome paired with the translatome to calculate translation efficiencies by normalization with the amount of mRNAs. A purification method of the total RNAs from the same cell origin regardless of ribosome association should be developed. Recently, Biever et al. combined microdissection of the neuropil (axon-enriched region) and somata (cell-body-enriched region) of the CA1 area of the hippocampus with polysome profiling and ribosome profiling, demonstrating that a specific subset of mRNAs are preferentially translated by monosomes in the neuropil, while somatic translation is mediated by polysomes [105]. As shown in their study, ribosome profiling can be exploited to uncover or explain subcellular controls of translation as well as cell-type specific ones, if the technique is combined with an appropriate isolation method.

### 5.3. Targeting Ribonucleoprotein Complexes

RNA binding proteins (RBPs) have drawn considerable interest in their roles for neural function that is evidenced by the association of a number of RBPs with neurological and neuropsychiatric diseases. RBPs are involved in many RNA processing events such as local transport, alternative splicing, and translation. Especially, given that translation itself is coordinated by huge ribonucleoprotein complexes that consist of many RNAs (rRNAs, tRNAs, mRNAs, and even long-noncoding RNAs) and RBPs (ribosomal proteins and translation regulatory proteins), elucidating the molecular actions of RBPs may be pivotal for understanding molecular mechanisms underlying any translation control.

Several RBPs such as FMRP and TDP-43 are known to be implicated in the translational regulation of selective transcripts in neuron-specific contexts, but most of their molecular behavior was initially studied by conventional methods targeting one or a few representative RNAs or proteins [106]. A key to explain the molecular basis of the gene-specific translational controls mediated by these RBPs is identifying their direct RNA targets. To this end, Darnell and his colleagues developed a powerful method termed HITS-CLIP (High-Throughput Sequencing of RNA isolated by CrossLinking ImmunoPrecipitation, also known as CLIP-seq) to investigate direct targets of a certain RBP at a genome-wide scale [107]. HITS-CLIP uses UV irradiation to fixate in vivo RBP-RNA interactions by inducing covalent bonds between the RBP and RNA that are within distances of a few angstroms. Unprotected RNA regions of the RBP-RNA complexes are removed by nuclease digestion and the complexes are isolated by the RBP-specific antibodies. The protected RNA fragments are purified and subjected to high-throughput sequencing to scrutinize the RBP-RNA interactions in vivo cellular environment. Since UV-crosslinking preserves only protein-RNA interaction within a few angstroms, not protein-protein interaction and enables stringent washing to avoid contamination with non-specific RNA, HITS-CLIP technique ensures fine identification of direct targets of RBPs unless the used antibody pulls down other non-specific RBPs.

The first application of HITS-CLIP determined RNA targets of Nova, a neuron-specific RBP that is associated with paraneoplastic neurologic degeneration, demonstrating the powerfulness of the technique to find RBP-RNA interaction sites in an unbiased and comprehensive manner [107]. The study performed by Licatalosi et al. provided a genome-wide map of Nova-RNA interaction by HITS-CLIP, confirming the previous prediction that the RNA binding site of Nova decides the patterns of alternative splicing and unveiling the RBP regulates alternative poly-adenylation by its binding to 3′ UTR. The robustness of the HITS-CLIP technique to define the precise location of RBP binding was potentiated by Cho et al.’s study on LIN28A, an embryonic stem cell-specific RBP. They determined the fine RBP binding sites at single-nucleotide resolution using the information of altered bases of HITS-CLIP sequencing tags that were induced by UV crosslinking [108]. HITS-CLIP was also exploited to study the mechanisms of translational regulation by FMRP [15]. Darnell et al. applied HITS-CLIP to FMRP, revealing that a significant portion of FMRP targets was implicated in synaptic plasticity and autism spectrum disorder. Unexpectedly, FMRP binding sites were broadly distributed across the coding sequences of target mRNAs, while many other RBPs that up- or down-regulate translation had been reported to bind to the specific spots within the UTRs of their targets. Based on these observations, the authors demonstrated that FMRP reversibly stalls ribosomes specifically on its target mRNAs, thereby suppressing translation.

Given that the interplay between RBP and RNA is important for neuronal functions such as RNA transport to and local translation at the synapses, the application of HITS-CLIP to neuronal RBPs will add insight in our understanding into molecular mechanisms of crucial post-transcriptional regulation to neurons. Indeed, HITS-CLIP and its variants such as PAR-CLIP (PhotoActivatable Ribonucleoside-enhanced CrossLinking and ImmunoPrecipitation) and i-CLIP (**i**ndividual-nucleotide resolution CrossLinking and ImmunoPrecipitation) have been applied to numerous neuronal RBPs, including Nova, TDP-43, FMRP, and hnRNPs [107,109,110,111,112]. Most of these studies focused on the role of the RBPs as regulators of alternative splicing. However, as shown in the FMRP HITS-CLIP study, HITS-CLIP and its variants may be helpful for identifying the targets of translational controls that are mediated by certain RBPs. Recently, another variant, hi-CLIP (RNA **h**ybrid and **i**ndividual-nucleotide resolution CrossLinking and ImmunoPrecipitation), developed by Sugimoto et al. enabled the identification of the secondary structures of target RNAs that are recognized by RBPs, another important information to comprehend molecular actions of RBPs [113]. Although this technique has not yet been exploited in a neural context, it will be useful to study neuronal RBPs that may selectively bind to double-strand or hairpin structures of target RNAs.

### 5.4. Targeting Ribosomes

Tissues consist of multiple types of cells and the patterns of gene expression vary among the cell types. This heterogeneity of gene expression within tissues suggests the necessity of genome-wide studies using homogeneous populations of cells. Recently developed single-cell RNA-seq (scRNA-seq) may provide a strong solution for this issue by revealing gene expression profiles with single-cell origins [114,115]. However, considerable concern remains that isolating single cells or homogeneous cell population often requires harsh and time-consuming treatments that may disturb genuine gene expression patterns in the in vivo tissue environment. Two groups attempted to address this challenge by using genetically modified mice expressing the epitope-tagged ribosomal protein in a cell-type specific manner [116,117]. The tagged ribosomes substitute for essential translational machinery in the specified cells, associating with and translating mRNAs. Thereby, affinity-purification of the epitope-tagged ribosomes allows for rapid and immediate isolation of ribosome-associated mRNAs that reflects the gene expression pattern of a certain type of cells. This technique was termed TRAP, an abbreviation for translating ribosome affinity purification. Heiman et al. generated TRAP mice expressing EGFP-tagged ribosomal protein L10a from Drd1a and Drd2 receptor gene, recovering ribosome-associated mRNAs from striatonigral and striatopallidal cells, respectively. The first and subsequent studies using TRAPs demonstrated the power of the technique by successfully representing the gene expression profiles from more than 20 different types of central nervous system cells. Another similar strategy to TRAP that is called RiboTag was developed by Sanz and colleagues. They constructed a genetically modified mouse line in which three HA tags coding sequences were inserted into the chromosomal locus of Rpl22, another ribosomal protein-coding gene. By using *Cre* recombination system, HA-tagged RPL22 can be expressed in a certain specified type of cells. They crossed the RiboTag mouse line with neuron-specific *Cre* mouse lines and isolated mRNAs that are associated with the HA-tagged ribosome by immuno-purification. The gene expression profiles recovered from the purified mRNAs revealed the reliability of RiboTag technique to isolate the mRNAs that are being translated by ribosomes in a cell type-specific manner.

Since the strategy of RiboTag uses *Cre* recombinase system, this technique was more versatile than the initial version of TRAP; RiboTag mouse line can be crossed with various *Cre* recombinase-expressing mouse line to isolate the ribosome-associated mRNAs from a wide range of different cell types. Indeed, RiboTag had been exploited for the genome-wide studies on other types of cells such as macrophage as well as neurons. Later, *Cre* recombinase dependent TRAP was developed by Ye at al. and the TRAP strategy exploited diverse epitopes such as phosphorylated RPS6, extending its use to studies on other cell types and a wider range of contexts as well [118,119]. Especially, these modified strategies of TRAP and RiboTag have been applied to study activity-dependent gene expression controls in neurons, which will be discussed in detail below.

Recently, Barna and her colleagues exploited a similar approach to TRAP and RiboTag to isolate functional ribosomes and the interacting proteins with the ribosomes in mouse embryonic stem cells [120]. Notably, they used FLAG-tagged RPL36 and RPS17 protein for affinity purification of active ribosomes instead of HA-tagged RPL22 since many RPL22 proteins were not found in the assembled ribosomes in mouse embryonic stem cells. This observation suggests a necessity of a parallel comparison when different ribosomal proteins are tagged in TRAP or RiboTag technique. For example, RPL22-based TRAP may less effectively detect actively-translated mRNAs than other ribosomal proteins-based ones, if not many RPL22s exist in functional ribosomes in neural tissues as well. Moreover, as Barna’s group did in mouse embryonic stem cells, it will be intriguing to dissect the composition of translational machineries in neurons that may be distinct from other cell types as well as embryonic stem cells.

## 6. Unbiased Translatome-Wide Approach to Bridge Translation and Activity: PhosphoTRAP and IEG-RiboTag

Genetic identification of neural population processing specific information is one of the fundamental challenges to understand the brain [121]. Pinpointing cell types susceptible to certain pathological conditions also provides insight into the onset and progress of neurological and psychiatric disorders [122,123,124,125]. Activity-dependent translational control offers an invaluable toolset to identify the molecular signatures of an active ensemble responding to a specific stimulus based on the principle of TRAP-seq [118,119].

Phosphorylation of RPS6, a component of the ribosome 40S subunit, has been known for several decades [126,127]. After being first discovered in the regenerative liver in vivo [128], phosphorylation of RPS6 has been reported under various stimulating conditions of cells, especially neurons [127,129]. In the C-terminus of RPS6, five serine residues of phosphorylation target sites are located: S235, S236, S240, S244, and S247. The phosphorylation status of these sites is determined by a partially sequential combination of biochemical interaction. The phosphorylation of S235 and S236 is mediated by the activation of several signal transducing kinase cascade including S6K1, protein kinase A (PKA), ribosomal S6 kinase (RSK), protein kinase C (PKC), protein kinase G (PKG), and death-associated protein kinase (DAPK). S240/244/247 is exclusively phosphorylated by S6K1. S247 can also be phosphorylated by CK1. The phosphorylation by S6K1 is elicited by the mTOR pathway which turns on upon neuronal activation, while other kinases targeting S235/236 can be activated upon a broader range of signaling pathways other than neuronal activation [38,127]. Therefore, phosphorylation of S240/244/247 of RPS6 can be considered as a surrogate marker of neuronal activation.

Based on the fact that phosphorylation of RPS6 reflects the recent activation status of a neuron, TRAP-seq targeting phospho-RPS6, called phosphoTRAP, has been utilized to profile molecular signatures of the activated neuronal population [118]. Phosphorylated RPS6 and the expression of cFos were highly colocalized in the cell responding to a variety of stimuli ranging from pharmacological stimulation, hydration state, feeding, light, and social cues in a wide range of brain regions such as the hippocampus and striatum as well as several hypothalamic nuclei. The sequencing of RNA co-precipitated with phospho-RPS6 was capable to suggest putative marker genes based on the enrichment in experimental conditions. This technique was further used to identify the molecular signature of the active ensemble in warm sensing [130,131], olfactory receptor responding to a specific odorant [132], and pheromone receptor responding to pup-derived cues [133].

Although several groups utilized phosphoTRAP, it is interesting that the temporal interval from the presentation of stimuli to the sampling of the target brain area varies between one to four hours. Considering the well-established dynamics of mRNA and protein abundance in immediate early gene expression, understanding temporal dynamics of post-translational modifications of RPS6 would facilitate the standardized application of phosphoTRAP. In addition, the establishment of a gene set that can serve as a quality control marker, surely enriched in activated neurons, would also help setting consensus on the scientific rigor in the field. This seemingly simple goal may not be straightforward as it seems, because bias in the translational profile of the ribosome harboring phosphorylated RPS6 has not been discovered yet to our knowledge [134].

Another TRAP-based approach to identify markers of the activated ensemble is achieved by combining *Cre* DNA recombinase-dependent TRAP and *Cre* driver line under the control of *cis*-element of Arc, an immediate early gene (IEG-RiboTag) [119,135]. The methodology to gain a genetic handle to the activated neural ensemble is expanding [136]. These techniques mainly exploit the promoter or *cis*-element of immediate early genes, such as cFos or Arc, to express molecular switch such as *Cre* DNA recombinase in order to restrict the expression of, for example, GFP-L10a only in the activated neuronal population.

A traditional approach to discover marker genes for a specific set of neurons has relied on serendipitous observation or heuristic gene-by-gene exploration, which was not efficient and not capable of discovering the optimum markers. As we summarized in this section, an unbiased experimental approach to identify molecular signatures utilizing translation machinery accelerated finding useful makers that define the neural population responding to a specific stimulus. Still, the genome-wide dataset obtained by phosphoTRAP or IEG-RiboTag does not guarantee the identification of useful markers, because these techniques distinguish the translatome of neurons inactive conditions from that in resting conditions rather than responsive neurons from non-responsive neurons. Therefore, subsequent histological validation to colocalize a candidate marker with the expression of the immediate early gene is often required. It is of importance that cell atlas, built by single-cell RNA sequencing, focusing specific brain region is increasingly available [137,138,139]. It is valuable to map the genome-wide dataset obtained by activity-dependent translatome-based approaches onto reference atlas built from single-cell RNA sequencing. This computation-assisted deconvolution will further facilitate the identification of marker genes for an active ensemble.

## 7. Conclusion and Outlook

So far, we have explored how neural activity can shape mRNA translation. Activity-dependent changes in gene expression are an essential feature of neurons, where translational control plays a critical role. Extensive studies have elucidated the role of activity-dependent translational regulation and its underlying mechanisms in physiological and pathological conditions, mainly focusing on a specific context, such as learning and memory. Yet, in vivo electrophysiological studies of awake behaving animals point out that the brain is such a chatty organ filled with frequent electric firings. In this regard, it is tempting to speculate that activity-dependent translational control is utilized, more often than we imagined, to maintain the normal functioning of neurons. This view may help in understanding why so many neurological diseases and psychiatric disorders are associated with activity-dependent translational control.

Luckily, we are living in an era of exploding scientific methods to explore both neuroscience and translation. Evolving repertoire of neurotools—such as optogenetics for manipulation of a selected population of neurons at a millisecond precision, real-time imaging of neural activity, parallel recording of thousands of neurons—allows interrogating the impact of neural activation on protein synthesis. In addition, the advance of the high-throughput sequencing technology enables us to study translational controls at a genome-wide scale in more extensive and unbiased manner. As we discussed earlier, some techniques such as ribosome profiling and TRAP/Ribo-tag will contribute to identifying unknown translational regulation of neuronal genes and other techniques such as HITS-CLIP will be helpful for delineating the molecular mechanisms underlying the regulation. Furthermore, a combination of some techniques such as ribosome profiling and TRAP/Ribo-tag will enable studies on translation controls in the more-specific context of the nervous system such as activity-dependent translation. Compared to the conventional approaches that examine single or a few numbers of molecules, this new methodology is expected to more rapidly accelerate our progress of understanding neuronal gene regulation that may be closely associated with neurological and neuropsychiatric disorders.

## Figures and Tables

**Figure 1 ijms-21-01592-f001:**
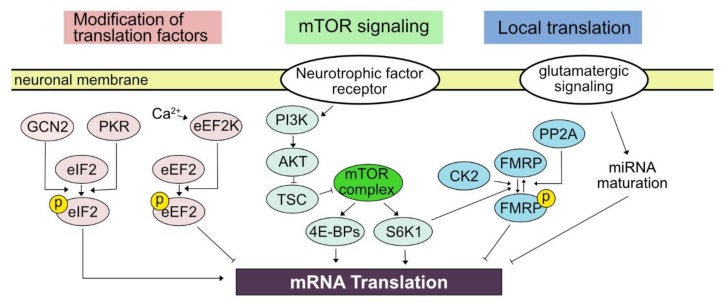
Cellular mechanisms mediating activity-dependent translational control. Intracellular signaling pathways that link neuronal activity to the regulation of mRNA translation can be classified into the modification of translation factors, mTOR signaling pathway, and local translational control. Phosphorylation of eIF2 by GCN2/PKR and phosphorylation of eEF2 by eEF2K either activates or represses mRNA translation, respectively. mTOR complex is activated by a series of signaling cascade comprised of PI3K-AKT-TSC and affects protein synthesis through 4E-BPs and S6K1. Local translational regulation involves reversible post-translational modifications of FMRP as well as the maturation of miRNA.

**Figure 2 ijms-21-01592-f002:**
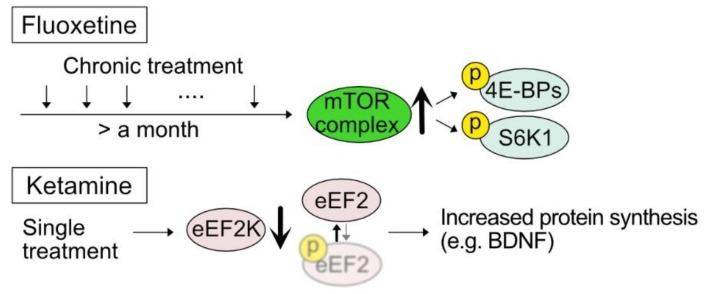
Translational control in antidepressant action. Antidepressant treatment regulates translation through the posttranslational modifications of the components in translation regulatory pathway. Chronic treatment of fluoxetine triggers the activation of the mTOR signaling pathway (upward arrow) leading to phosphorylation of 4E-BPs and S6K1. A single treatment of a subanesthetic dose of ketamine suppresses eEF2K activity (downward arrow) to increase protein synthesis, including BDNF, through dephosphorylation of eEF2.

**Figure 3 ijms-21-01592-f003:**
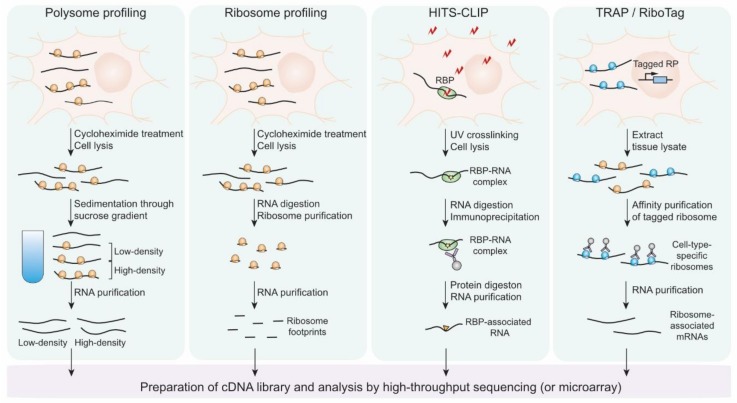
Schematic presentation of genome-wide methods to study translational controls in neurons. The workflows of polysome profiling, ribosome profiling, HITS-CLIP and TRAP/RiboTag (from left to right panel in the figure).

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
