# Peer review of "Comprehensive Genome-Wide Approaches to Activity-Dependent Translational Control in Neurons"

_ijms, 2020, doi:10.3390/ijms21051592_

Round 1
Reviewer 1 Report
This revision, by Choe and Cho, describes a complete and comprehensive review of the latest genomic approaches, regarding the genetic variants involved in neuronal disorders, focused on the implication of translational control for neuronal gene regulation and functions in the brain.
Minor comments
- Line 84. Missing references related to ”…reports over the past decades”
- Lines 369-370. Please mention some of the conventional methods used
- It will be extremely helpful to have a cartoon or table linking the conventional techniques to the newest approaches described in this review as a summary.
Reviewer 2 Report
Comments
The review submitted by Choe and Cho summarizes the current genome-wide techniques to analyze translational regulation in neurons. It especially focuses on the regulation induced upon neuronal activity.
The paper is unevenly written. There are sections very clear and well written, while others have incorrect singular/plural concordance, linking words are missing or are unnecessary long. There are several sentences that need to be improved (g. lines 191, 197, 199, 201,217, 244-247, 321, 330, 333, 340, 395-407, 471…). Even the title, should be “Comprehensive genome-wide approaches to activity dependent translational control in neurons”. I would not say that Autism spectrum disorder is a “psychiatric disorder” (in abstract and main text). I suggest to change it to “neurodevelopmental disorder”. Within the section “Targeting ribonucleoprotein complexes”, it should be stated the meaning of acronyms HITS-CLIP, PAR-CLIP, i-CLIP. Moreover, hi-CLIP is missing (Sugimoto et al, 2017, Nature Protocols). This technique is relevant as it allows the identification of RNA duplexes that are bound to RBPs. For certain proteins, such as double-stranded RBPs, this is the preferred technique. In the section “Targeting Ribosomes”, authors should also include the study by Simsek et al., 2017, Cell. They were able to isolate functional ribosomes, and show the heterogenity of ribosome associated proteins. During the reviewing process of this manuscript, a new screening combining microdissection of the Neuropil-soma areas of the CA1 area of the hippocampus, polysome profiling and Riboseq has been published (Biever et al, 2020, Science). The authors show that there are mRNAs that are preferentially transcribed by monosomes in the neuropil, while somatic translation is mediated by polysomes. The review would be benefited by including this additional level of translational regulation in neurons. Line 94: it should be “immediate early gene response”. Line 157: change the reference Buffinton et al, 2014 by its corresponding number. Line 296: it should be stated that DSM-5 corresponds to the diagnostic criteria for ASD, and a reference or link should be provided Line 357: AMRA should be AMPA
